# Open Temporal Relation Extraction for Question Answering

**Chao Shang**                         CHAO.SHANG3@JD.COM
**Peng Qi**                           PENG.QI@JD.COM
**Guangtao Wang**                   GUANGTAO.WANG@JD.COM
**Jing Huang**                        JING.HUANG@JD.COM
**Youzheng Wu**                     WUYOUZHENG1@JD.COM
**Bowen Zhou**                      BOWEN.ZHOU@JD.COM
*JD AI Research, Mountain View, CA 94043*

## Abstract

Understanding the temporal relations among events in text is a critical aspect of reading comprehension, which can be evaluated in the form of temporal question answering (TQA). When explicit timestamps are absent, TQA is a challenging task that requires models to understand the nuanced difference in textual expressions that indicate different temporal relations (e.g., "What happened right before dawn" indicates a small subset of "What happened before dawn"). In this paper, we propose to reformulate the task of TQA as open temporal relation extraction. Specifically, we decompose each question into a question event (e.g., "dawn") and an open temporal relation (OTR, e.g., "happened before") which is not pre-defined nor with timestamps, and ground the former in the context while sharing the representation of the latter across contexts. This OTR for QA formulation has two advantages: 1) it allows us to learn context-agnostic, free-text-based relation representations that generalize across different contexts and events, which leads to higher data efficiency; 2) it allows us to explicitly model the differences in temporal relations with a contrastive loss function, which helps better capture mutually exclusive relations (e.g., an event cannot simultaneously "happen before" and "happen after" another) as well as more nuanced differences (e.g., not everything that "happened before" an event "happened right before" it). Empirical evaluations on the TORQUE challenge, a recently released dataset for temporal ordering questions, show that our approach attains significant improvements correspondingly over the state of the art performance, especially gains more on EM consistency computed on the contrast question sets.

## 1. Introduction

Despite the significant progress made in question answering (QA) over the recent years [Gupta and Gupta, 2012, Mishra and Jain, 2016, Höffner et al., 2017], questions that involve temporal reasoning between events in text have received relatively little attention. Previous work has shown that many existing QA techniques are usually ill-equipped to tackle the problem of temporal reasoning [Zhou et al., 2019, Ning et al., 2020]. This failure is partly due to the insensitivity of these QA models to nuanced and subtle textual changes in temporal questions that could imply completely different relations. For instance, Figure 1, "what happened right after the election", "what failed to happen after the election", and "what happened right before the election" can have mutually exclusive sets of answer events given the same context.

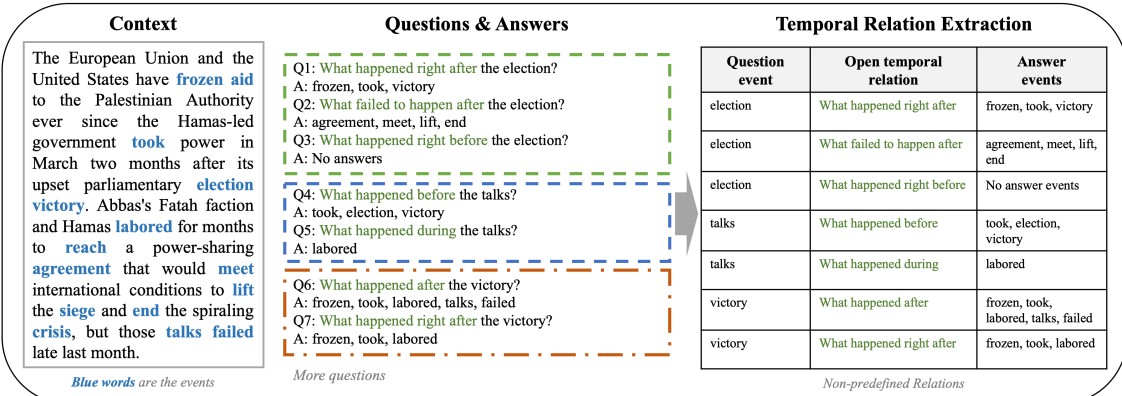

Figure 1: Examples of temporal question answering and its reformulation to open temporal relation extraction.

Most previous approaches towards temporal question answering focus on settings where explicit timestamps (e.g., "August 2020") are available in the question, the context, and/or knowledge bases [Jin et al., 2020, Harabagiu and Bejan, 2005, Jia et al., 2018b,a]. However, explicit timestamps are not available in many texts, where it is still important to understand the temporal relation between events (e.g., novels, news reports). In these cases, one would need to seek clues from the context in which events are described to understand the relative temporal ordering of events to answer questions (see Figure 1 for an example). Consequently, this task is more taxing for human annotators which limits the amount of annotated data available. On the other hand, the potential set of questions asked is also large thanks to its text-based nature, which is further exacerbated by the tight coupling between questions and the context. Both of these characteristics render this task difficult for powerful one-size-fits-all neural network models since these models are prone to overfitting.

In this paper, we propose to reformulate the problem of temporal question answering as one of open temporal relation extraction. Specifically, we decompose each question into an *open temporal relation* (OTR) expressed in natural language (similar to that of open information extraction [Etzioni et al., 2008]), as well as a *question event* that is context-dependent. This formulation (OTR-QA) has two advantages. Firstly, it allows us to model temporal relations in a context-agnostic manner, which shares supervision signal from different contexts and events to the same underlying open temporal relation. As a result, OTR-QA is much more data-efficient compared to its BERT-based counterpart that does not explicitly consider this decomposition, and generalizes better with the same amount of training data. We demonstrate that OTR-QA significantly improves upon the previous state of the art on the TORQUE [Ning et al., 2020] dataset. Secondly, this reformulation allows us to explicitly model the differences in temporal relations with a contrastive loss function, which helps capture mutually exclusive relations (e.g., Q1, Q2, and Q3 in Figure 1) and relations whose differences are more nuanced (e.g., Q6 and Q7 in Figure 1 , where the answer set of Q7 is a subset of that of Q6). This results in much more coherent answers across questions that are within *contrast groups* defined in TORQUE, which are usually

questions centered around the same question event but have different answers given the same context (e.g., the green and blue dashed boxes in Figure 1 is each a contrast group).

To recap, our main contributions in this paper are: 1) we propose to reformulate temporal question answering as a new task, open temporal relation extraction, which is more data-efficient compared to its counterparts that do not factorize the problem; 2) we further show that this formulation allows us to explicitly model contrasting temporal relations with a contrastive loss function, which further improves the coherence of model prediction; 3) our proposed model outperforms the previous state of the art on the TORQUE dataset by a large margin.

## 2. Related Works

Time information is crucial for understanding the events and temporal relations among events. Existing studies have explored the time information from different perspectives. Some work made use of the duration and frequency of events for temporal commonsense reasoning [Vempala et al., 2018, Zhou et al., 2019, 2020]; some work focused on temporal relation extraction [Ning et al., 2019a,b], timeline construction [Leeuwenberg and Moens, 2018] and temporal question answering [Ning et al., 2020]. Recent work DEER [Han et al., 2020] took the pre-training approach that trained a language model to focus on event temporal relations. A large amount of training samples were created to simulate the QA and information extraction tasks for event temporal understanding.

**Temporal Relation Types.** The attempts on temporal relation extraction are usually making pairwise decisions between each pair of events given a pre-defined temporal relation. Existing approaches for temporal processing often used the interval representation of events proposed in Allen [1984] which includes 13 relation types in total. When a relation is not clear, a vague or none relation could be also added as another relation type. In CAEVO [Chambers et al., 2014], the defined temporal relation types are {before, after, includes, is included, equal, vague}. These approaches all used a predefined set of relation types. However, in the temporal question answering task, the temporal relations included in questions could be diverse that cannot be specified in advance. In this paper, we design the model to handle the open temporal relations without defining the pre-defined types.

**Temporal Question Answering (TQA).** Most of TQA work required timestamps in the input. For example ForecastQA [Jin et al., 2020] formulates the forecasting problem as a multiple-choice question answering task, where both the articles and questions include the timestamps. It introduces a timestamp constraint per question that prohibits the model from accessing articles published after the timestamp. Another task is temporal question answering over knowledge bases (KB) [Jia et al., 2018b,a], which retrieves time information from the KB. Few work has been done to explore the temporal relations without timestamps. The recent released TORQUE [Ning et al., 2020] is a designed dataset that explores the temporal ordering relations between events described in a passage of text. However, understanding these temporal relations is challenging as they cannot be pre-defined in advance, and is also sensitive to the small changes on implicit temporal keywords. To handle these challenges, we propose the OTR-QA model in this paper. What's more, there are few approaches that attempt to train the model with closely related questions, which drawing the ideas from the contrastive learning [Oord et al., 2018]. Both [Asai and Hajishirzi, 2020]

and [Dua et al., 2021] take advantage of the closely related questions to figure out the difference between the inputs that leads to the expected difference between their answer. Our OTR-QA also captures the temporal difference between the related temporal questions by proposing a contrastive loss function.

## 3. Methodology

In this section, we first formulate the open temporal relation extraction problem and then introduce our model that builds on this formulation for temporal question answering.

### 3.1 Problem Definition

**Temporal Order Question Answering**. The task of temporal order question answering (TORQUE) [Ning et al., 2020] is a question answering task given a paragraph of text as context. Formally, let $C = \{s_1, \cdots, s_n\}$ be a piece of text with $n$ sentences as context, a set of $m$ events, $E = \{e_1, \cdots, e_m\}$ are defined, which covers salient verbs (such as "investigating" and "said") and nouns (such as "accident" and "landslide"). The task of TORQUE aims at answering questions $Q$ that concern with the temporal order of these events where the answer event set $A_e \subset E$ (see Figure 1 for examples).

Semantically, questions in TORQUE largely take the form of expressing the temporal relation of events in $C$ with respect to a question event $e_q \in E$. For instance, the first question in Figure 1 queries for events that "happened right after" the event "election", which is one of the highlighted events in the context. Since these questions are written in natural language, they can express nuanced temporal relations such as "happened right after", "might happen after", or "might have happened before", which is difficult to enumerate or define ahead of time. We therefore treat these as **open temporal relations** (OTR), and propose to answer these questions by extracting these OTRs.

**Open Temporal Relation Extraction**. Given a question $Q$ based on context $C$ with event set $E$, we begin by decomposing $Q$ into the open temporal relation $r$ it expresses, and the question event $e_q \in E$. The task of open temporal relation extraction (OTRE) is then to find all candidate events $e_a \in E$ for which the temporal relation triple $(e_q, r, e_a)$ holds. Once this is done, we have recovered the desired answer set $A_e = \{e_a | e_a \in E \text{ and } (e_q, r, e_a) \in \text{TemporalRelations}(C)\}$.[1]

This new task formulation has two advantages. 1) Instead of treating the question as a whole for answer event prediction, which is standard for existing question answering models, OTRE's explicit decomposition of the question provides an opportunity to construct a context-agnostic representation of the temporal relations between events, which is more effective at leveraging the limited supervision signal available to generalize to unseen contexts and events. 2) It also allows us to more effectively and explicitly model the difference in temporal relations that are expressed similarly in natural language with a contrastive loss function, including ones that are mutually exclusive (e.g., "happened before" and "happened after") and ones that are different in a more nuanced manner (e.g., "happened before" and "happened right before"). This helps improve the model's coherence in its answers to

---

1. Note that TemporalRelations($C$) does not explicitly exist, and is used here for notational purposes only. The task of open temporal relation extraction itself is to learn to find such triples from annotated question answering data.

questions that are centered around the same question event. The OTRE decomposition allows us to propagate the difference defined by different answer sets effectively to open temporal relation representations, which further helps with generalization.

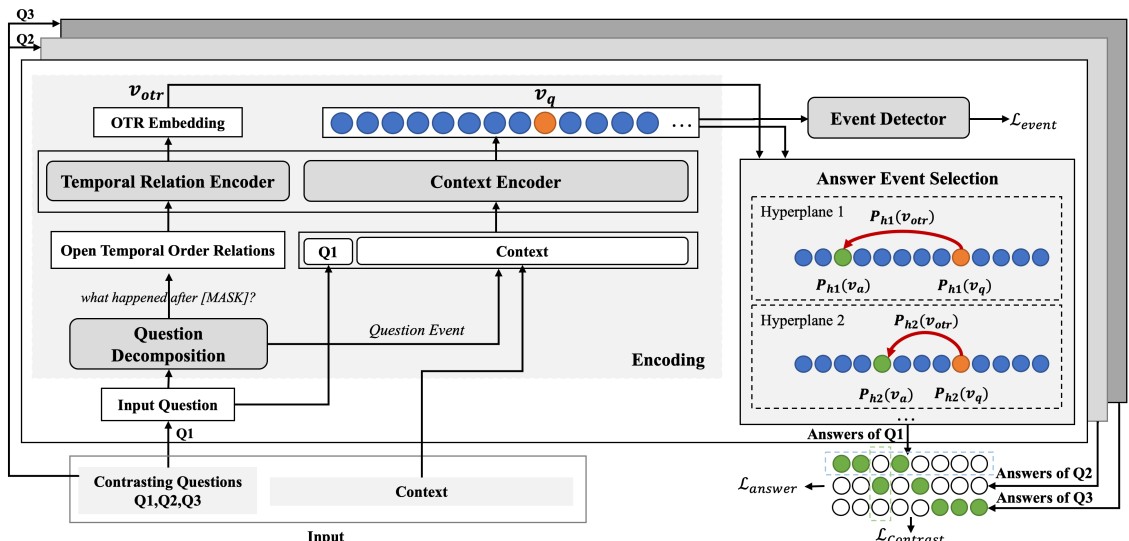

Figure 2: The architecture of our OTR-QA model.

## 3.2 The OTR-QA Model

In this section, we introduce our question answering model based on open temporal relation extraction, OTR-QA. Given an input question $Q$, it is mainly processed in two steps to arrive at the final answer: 1) Question decomposition and context encoder; 2) Answer event prediction from these learned representations that identifies temporal relation triples from the context. During training, we further regularize the model with two auxiliary loss functions: one for event detection, and the other for contrasting questions that share the same $e_q$ but have different answers.

### 3.2.1 Question Decomposition and Text Encoder

**Question Decomposition.** To decompose each question into a question event $e_q$ and an open temporal relation $r$, we look for event-like words in the question. As the event set $E$ is provided during training, we extract nouns and verbs from the question which also appear in context as the question event. Since TORQUE features "warm-up" questions that do not have explicit event mentions (e.g., "What has already happened" and "What will happen in the future"), we define a special event "current time" as $e_q$, which represents the event at current point-in-time, when an event in $E$ cannot be found in the question.

Once we obtain the question event $e_q$, we define the rest of the question as the open temporal relation $r$ by removing i) the event words in $e_q$, ii) stop words and iii) low frequency words (based on statistical analysis on all training data).

**Temporal Relation Encoder.** We replace removed words in the question with a special "[MASK]" token as placeholders before feeding it into a pretrained language model (such as BERT). We average pool the resulting representations over the entire sequence as the representation of the OTR, $\boldsymbol{v}_r$.

**Context Encoder.** Similarly, we encode the context with a pretrained langauge model to get the word sequence embeddings for open temporal relation extraction. For each question, we first concatenate it with the context (that always consists of two sentences in TORQUE) as "[CLS] + question + [SEP] + sentence1 [SEP] sentence2 [SEP]" as input to the language model. If a word consists of multiple pretrained LM tokens, we make use the embedding of the first wordpiece as the to represent the word. We then extract i) the question event representation $\boldsymbol{v}_q$ by averaging the embeddings of words in question event $e_q$, which could contain multiple words; ii) the context word sequence embeddings $\boldsymbol{V} = [\boldsymbol{v}_0, \boldsymbol{v}_1, \cdots, \boldsymbol{v}_n]$ for further processing. Note that we share the same pre-trained language model for both temporal relation encoder and context encoder to prevent overfitting and reduce the parameter budget of the entire model.

### 3.2.2 Multi-task Learning

**Answer Event Prediction.** Since we reformulate the answer event prediction as a triple fact prediction task via open temporal order relation extraction, we are interested in defining a scoring function for each candidate triple $(e_q, r, e_a)$. Let $\boldsymbol{v}_q \in \mathcal{R}^d$ and $\boldsymbol{v}_r \in \mathcal{R}^d$ be the representation of question event $e_q$ and OTR $r$, respectively, and $E = \{e_1, \cdots, e_m\}$ be the set of $m$ candidate answer events, and $\boldsymbol{v}_i \in \mathcal{R}^d$ be the representation of $i$-th event, and $d$ is the output dimension of pretrained LM. Then, we can construct $m$ triples as $\mathcal{T} = \{(e_q, r, e_i)|e_i \in E\}$ $(1 \leq i \leq m)$, and view answer event prediction as a task to identify the true triple fact in $\mathcal{T}$.

For a given question event $e_q$ and OTR $r$, there could be multiple answer events. That is, there could be multiple true triple facts in $\mathcal{T}$. Inspired by TransH [Wang et al., 2014] in handling 1-to-N relations and HyTE [Dasgupta et al., 2018] in predicting temporal scopes, we propose a multiple hyperplane projection with top-$k$ selection method based score function.

We first define $K$ hyperplanes as $\boldsymbol{W}_{\text{hypeplane}} \in \mathcal{R}^{K \times d}$. For a given OTR representation $\boldsymbol{v}_r$, we first identify the top $k$ related hyperplanes as

$$\boldsymbol{W}_h = [\boldsymbol{W}_{h,1}, \cdots, \boldsymbol{W}_{h,k}] = \text{topk}(\text{softmax}(\boldsymbol{W}_{\text{hypeplane}} \boldsymbol{v}_r)) \in \mathcal{R}^{k \times d} \quad (k \leq K) \qquad (1)$$

Then, for each hyperplane $\boldsymbol{W}_{h,i}$, we project the events and relations on this hyperplane by

$$P_{h_i}(v_q) = \boldsymbol{v}_q - (\boldsymbol{W}_{h,i}^T \boldsymbol{v}_q)\boldsymbol{W}_{h,i}, P_{h_i}(v_j) = \boldsymbol{v}_j - (\boldsymbol{W}_{h,i}^T \boldsymbol{v}_j)\boldsymbol{W}_{h,i}, P_{h_i}(v_r) = \boldsymbol{v}_r - (\boldsymbol{W}_{h,i}^T \boldsymbol{v}_r)\boldsymbol{W}_{h,i} \ (2)$$

Afterwards, based on the translation score function [Feng et al., 2016], which aims at keeping the same direction between the two representations, we define a triple score function on $(e_q, r, e_j)$ as follows

$$f_{h_i}(e_q, r, e_j) = (P_{h_i}(\boldsymbol{v}_q)) + P_{h_i}(\boldsymbol{v}_r))^T P_{h_i}(\boldsymbol{v}_j)) + (P_{h_i}(\boldsymbol{v}_j) - P_{h_i}(\boldsymbol{v}_r))^T P_{h_i}(\boldsymbol{v}_q) \qquad (3)$$

As we have $k$ hyperplanes, by taking the adaptive combination of multiple hyperplanes, our final triple score function is defined as

$$f(e_q, r, v_j) = \sum_{i=1}^{k} \boldsymbol{\theta}_i f_{h_i}(e_q, r, v_j) \tag{4}$$

where $\boldsymbol{\theta}_i$ is the parameter of the neural network.

Let $\boldsymbol{y}_i = [y_{i,0}, \cdots, y_{i,m}]$ be the true answer event label sequence w.r.t. question $i$ on context sentences, where $y_{i,j} \in \{0, 1\}$, and $y_{i,j} = 1$ represents $j$-th word is an answer event w.r.t. question $i$. Here we minimize the binary cross-entropy loss function[2]

$$\mathcal{L}_{\text{answer}} = -\frac{1}{m} \sum_{i=1}^{m} y_i \log(\text{sigmoid}(f_{\boldsymbol{v}_r}(\boldsymbol{v}_q, \boldsymbol{v}_a))) + (1 - y_i) \log(1 - \text{sigmoid}(f_{\boldsymbol{v}_r}(\boldsymbol{v}_q, \boldsymbol{v}_a))) \tag{5}$$

**Event Detector.** The event detector takes the word embeddings $\boldsymbol{V} = [\boldsymbol{v}_0, \boldsymbol{v}_1, \cdots, \boldsymbol{v}_n]$ as input, and views event detection as a binary classification task, where the true event label sequence is $\tilde{\boldsymbol{y}} = [\tilde{y}_0, \cdots, \tilde{y}_n]$, and $\tilde{y}_i = 1$ means that the $i$-th word is an event and $\tilde{y}_i = 0$ means $i$-th word is non-event. Let $\boldsymbol{Pr} = [p_0, \cdots, p_n]$ be the output of logistic regression model over each word embedding in $\boldsymbol{V}$, which indicates the probability of each word being an event word, the event detection task aims to optimize the following loss function

$$\mathcal{L}_{\text{event}} = -\frac{1}{n} \sum_{i=1}^{n} \tilde{y}_i \log(p_i) + (1 - \tilde{y}_i) \log(1 - p_i) \tag{6}$$

**Contrastive Loss.** Let $C$ be the number of contrastive questions. For a given question $i$ $(1 \le i \le C)$, we could get the scores of all triples via function 4 as $\boldsymbol{S}_i = [\boldsymbol{s}_{i,0}, \cdots, \boldsymbol{s}_{i,m}]$, here $\boldsymbol{s}_{i,j}$ is the score for word $j$ under question $i$. By concatenating scores of $C$ questions together, we can get a score tensor as $\boldsymbol{S} = [\boldsymbol{S}_1; \cdots; \boldsymbol{S}_C] \in \mathcal{R}^{C \times m}$. Then, we can get the probability of each word being an event as follows

$$\boldsymbol{p}_{i,j} = \frac{\exp(\boldsymbol{S}_{ij})}{\sum_{l=1}^{C} \exp(\boldsymbol{S}_{il})} \tag{7}$$

where $\boldsymbol{p}_{i,j}$ represents the probability of word $j$ being an answer event w.r.t. question $i$ $(1 \le i \le C, 1 \le j \le m)$.

For each word, we use the $\hat{\boldsymbol{y}}_j = [y_{j,0}, \cdots, y_{j,C}]$ as label for word $j$. Then, the contrastive constraint aims to optimize the following loss function

$$\mathcal{L}_{\text{contrast}} = -\frac{1}{m} \sum_{j=0}^{m} o_j \hat{\boldsymbol{y}}_j log(\boldsymbol{p}_j) \tag{8}$$

where $O = \{o_0, \cdots, o_m\}$ is a binary event indicator (1 or 0) and $\hat{\boldsymbol{y}}_j$ is a one-hot encoding.
**Multi-task Training.** Finally, we propose the combined loss function

$$\mathcal{L} = \mathcal{L}_{\text{answer}} + \alpha \mathcal{L}_{\text{event}} + \beta \mathcal{L}_{\text{contrast}}, \tag{9}$$

where $\alpha \ge 0$ and $\beta \ge 0$ are the hyper-parameters.

---

2. For simplicity, in our model implementation we consider all context words candidate "events" for answer prediction loss computation. We introduce the event detection regularization to help the pretrained-language model learn to distinguish events from non-events.

## 4. Experiments

In this section we validate our proposed OTR for TQA formulation on the recently released dataset TORQUE [3]. Our experimental results show that our approach obtains significant improvements over the baseline models in [Ning et al., 2020].

### 4.1 Experimental Setup

**Data.** TORQUE is a temporal ordering question answering dataset. It built on 3.2k news snippets with 25k events and 21k user-generated and fully answered temporal order relation questions. All the questions are used to query temporal relations between events in the context, which consists of two sentences. This task also provides contrast questions which slightly modify the original questions, but dramatically change the answers. Events were defined as either a verb or a noun. Note that over 95% of questions could be captured by an "(event, relation, ?)" using the question decomposition from section 3.2.1.

**Evaluation Metrics.** The evaluation metrics are the same in [Ning et al., 2020], which include the standard macro F1, exact-match (EM) and EM consistency (C), which is the percentage of contrast groups for which a model's predictions have $F1 \geq 80\%$ for all questions in a group.

**Baselines.** The baseline model is a neural reading comprehension model from [Ning et al., 2020]. The input to the baseline is in the format of "[CLS] + question + [SEP] + context [SEP]", which is fed to a pretrained language models (BERT or RoBERTa). The output of each word in the context from the encoder is binary classified either as answer to the question or not. Note that we do not include the model and results from DEER [Han et al., 2020] because it used different training data in addition to TORQUE. DEER not only uses TORQUE data but also takes extra 10 million sentence passages for training the model. Even though DEER uses more data, the improvement is still similar to us.

**Training hyper-parameters.** We use the following hyper-parameters for training the OTR-QA models: learning rate $\{1e^{-5}, 5e^{-6}, 1e^{-6}\}$, $\alpha$ $\{0.5, 0.6, 0.7, 0.8, 0.9\}$ and $\beta$ $\{0.5, 0.6, 0.7, 0.8, 0.9\}$. We set dropout ratio to 0.2, the $k$ of top-k function to 3. Our models are implemented by PyTorch and trained using NVIDIA Tesla V100 GPUs.

### 4.2 Experimental Results

**Main Results.** Table 1 compares our proposed OTR-QA with baselines in terms of F1, EM and C, over validation and test data sets, respectively. We first observe that, our proposed OTR-QA model + RoBERTa-large has achieved the state-of-the-art performance in terms of all metrics on both validation and test data sets. This implies that our OTR-QA model is superior to baselines on open temporal relation learning and further better performance on answer event prediction. In addition, our method obtains 3% absolute gain on the C score over the corresponding baseline model due to the contrastive loss used for training.

**Data Efficiency.** In Table 2 we compare our models trained from different amounts of training data with the corresponding baselines trained from the same data. First, we observe that when decreasing the amount of training data from 50% to 10%, the performance of the baseline model drops more than 10% in F1, while our OTR-QA models drops about 4%,

---

3. https://leaderboard.allenai.org/torque/

| Model | Val | | | Test | | |
|---|---|---|---|---|---|---|
| | F1 | EM | C | F1 | EM | C |
| BERT-base [Ning et al., 2020] | 67.6 | 39.6 | 24.3 | 67.2 | 39.8 | 23.6 |
| BERT-large [Ning et al., 2020] | 72.8 | 46.0 | 30.7 | 71.9 | 45.9 | 29.1 |
| RoBERTa-base [Ning et al., 2020] | 72.2 | 44.5 | 28.7 | 72.6 | 45.7 | 29.9 |
| RoBERTa-large [Ning et al., 2020] | 75.7 | 50.4 | 36.0 | 75.2 | 51.1 | 34.5 |
| OTR-QA (RoBERTa-base) | 75.2 | 49.2 | 36.1 | 73.4 | 47.1 | 32.7 |
| OTR-QA (RoBERTa-large) | **77.1** | **51.6** | **40.6** | **76.3** | **52.6** | **37.1** |

Table 1: Results on TORQUE dev set and test set.

| Model | F1 | EM | C |
|---|---|---|---|
| RoBERTa-base (10% training data) | 57.3 | 33.3 | 13.8 |
| RoBERTa-base (20% training data) | 66.8 | 39.8 | 24.1 |
| RoBERTa-base (50% training data) | 69.7 | 44.3 | 27.8 |
| OTR-QA (RoBERTa-base) (10% training data) | 69.0 | 40.7 | 25.0 |
| OTR-QA (RoBERTa-base) (20% training data) | 71.2 | 43.3 | 29.1 |
| OTR-QA (RoBERTa-base) (50% training data) | 73.4 | 47.2 | 32.4 |

Table 2: Comparison of the baseline models and OTR-QA models trained using different percentages of training data. Results are on the validation data.

less affected by the data amount; second, the OTR-QA model trained with 20% of training data is better than the baseline model trained with 50% of training data. This proves that the OTR-QA is much more data-efficient, because it is designed to learn temporal relations in a context-agnostic manner, and thus needs less training data.

### 4.3 Ablation Study

We conduct the following ablation study to show the advantages of our OTR-QA model: first, we prepare a set of pre-defined temporal relation types and map each question to one of the types. For example, the type for "what happened before" is mapped to the type "before". We use temporal keywords, such as the "before", "after", "while", to define six temporal types. In Table 3, we observe that the performance of this setting is significantly worse than our OTR-QA model. By using "open" relation extraction without pre-defined relation types, our OTR-QA model is much better in learning the variations and subtleties of temporal relations from limited textual descriptions.

We also conduct ablation study on the loss functions used in the training by removing the event detection loss and contrastive loss separately. Performances on both settings have decreased. When removing the contrastive loss, the C score has a significant drop which proves the benefit of using this loss to handle small changes in different question. In addition, we implement a scoring function with just one hyperplane. We observe that our multi-hyperplane score has better performance than a single-hyperplane scoring function, which proves that mapping events into multi-hyperplane benefits the TQA task. We also

| Model | F1 | EM | C |
|---|---|---|---|
| OTR-QA (RoBERTa-base) | 75.2 | 49.2 | 36.1 |
| 1. replace OTR by predefined relation types | 71.7 | 45.0 | 29.1 |
| 2. remove event detection loss | 74.2 | 48.8 | 33.5 |
| 3. remove contrastive loss | 73.6 | 46.7 | 31.8 |
| 4. use single hyperplane score function | 74.5 | 49.1 | 32.8 |

Table 3: Experimental results of ablation tests on validation data.

notice that all these four settings have better performance than baseline model. Thus the OTR reformulation for the TQA task makes the main contribution of our approach.

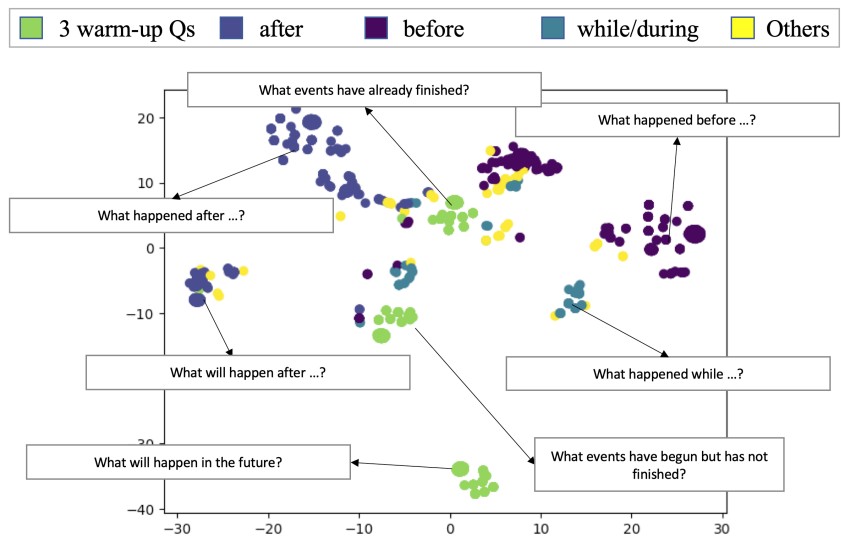

Figure 3: The t-SNE plot of the OTR embeddings learned from the OTR-QA model.

### 4.4 Analysis

We visualize the embeddings of the OTRs learned from our OTR-QA model by t-SNE [Van der Maaten and Hinton, 2008] plots in Figure 3. Our purpose is to show OTR-QA could learn better embeddings of open temporal relations. Here a better representation means that the similar relations are grouped together tightly and different relations are well separated. We first define five relation types so that we can check whether data points from the same type are clustered together. These five relation types are: three warm-up questions, before, after, while/during, and others. From Figure 3(b), we observe that the OTR embeddings from the OTR-QA model are well grouped by the same types. We further mark multiple OTR examples in Figure 3 (b) to illustrate the well-separated OTRs in the 2-D space.

## 5. Conclusion

In this paper we propose to reformulate the problem of temporal question answering as the open temporal relation (OTR) extraction. This new formulation (OTR-QA) has two advantages: first it models temporal relations in a context-agnostic manner, which shares learning signal from different contexts and events to the same underlying open temporal relation. As a result, OTR-QA is much more data-efficient compared to its counterpart that does not use this formulation, and generalizes better with the same amount of training data; Second this reformulation allows us to model the differences in temporal relations with a contrastive loss function, which helps discriminating mutually exclusive relations and improve EM consistency score. We demonstrate that our OTR-QA model significantly improves upon the previous methods on TORQUE, which further proves the efficiency and efficacy of our approach.

## Acknowledgments

This work was supported by the National Key R&D Program of China under Grant No. 2020AAA0108600.

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
