# OpenReview forum: "Open Temporal Relation Extraction for Question Answering"
_AKBC.ws/2021/Conference — AKBC 2021_

### Official Review · Reviewer_2dLF · 2021-07-20

**Rating:** 8
**Confidence:** 4

**Review:**

This paper reformulates TORQUE questions (temporal-ordering-focused reading comprehension questions)
as open temporal relation extraction.  Questions are decomposed as (source event, relation, ?)
queries, where the ? corresponds to another event in the passage.  After this decomposition, events
from the passage and the extracted relation are modeled in a way similar to knowledge base
completion, using multiple hyperplane projections.  An additional contrastive loss is added, which
basically amounts to normalizing event answer scores across contrasting questions, when it is known
a priori that an event has exactly one answer among the set of questions.  Experimental results show
that this model outperforms prior work on TORQUE, and good ablations are done to show the impact of
various modeling decisions that were made.

Overall, I think the paper presents a straightforward but novel idea, executes this idea well, and
evaluates it thoroughly.  I think the ideas and experiments here are interesting and timely.  The
results with a contrastive loss and open relation modeling add to the body of evidence of the
utility of these methods generally.  There are a few things that I would recommend improving in the
paper, but it's already useful as it is.


More detailed comments:

Related work on contrastive losses is missing.  Asai and Hajishirzi, 2020, "Logic- guided data
augmentation and regularization for consistent question answering", is a good starting place for
finding related work in this area.  The question-conditional contrastive estimation method from
Dheeru Dua et al. 2021, "Learning with Instance Bundles for Reading Comprehension" seems like it is
basically identical to the loss presented in this paper (this second one is a recent enough paper
that a comparison is not expected, and a missing citation is not surprising, I'm just providing it
FYI).

Do you know what percentage of the training data is not well captured by an (event, relation, ?)
query?  That is, how often does your decomposition fail?

3.2.1 - it's not clear what exactly is being averaged to compute v_q - all of the words in the
question?  Just the event word in the question?  The event word in the question and the
corresponding event word(s) in the passage?

Using a fixed set of temporal relations is a nice ablation, I like including that experiment quite a
bit.

Section 4.4 - t-SNE is generally not very informative, and I disagree with the text in this section
that says that the OTR model has a better clustering than the baseline.  It does not seem nearly so
clear cut to me as is stated in the text.  I think this section would be better to just be removed
from the paper.  It doesn't really add anything that's scientifically useful.

---

> ### Author Response · Authors · 2021-07-28
> **Thanks for your review and thoughtful suggestions!**
>
> We thank the reviewer for the excellent summary of our paper, and will update the paper based on your suggestions. In what follows we respond to the questions one by one.
>
> Q1: Related work on contrastive losses is missing.
>
> Response: We will add more related papers about contrastive learning including the papers you suggested. These contrastive methods learn representations by contrasting positive and negative examples, which have been used in both computer vision and NLP tasks.
> Some related papers that we will add to our references are:
>
> [1] Logic-Guided Data Augmentation and Regularization for Consistent Question Answering
>
> [2] Learning with Instance Bundles for Reading Comprehension
>
> [3] Contrastive estimation: training log-linear models on unlabeled data
>
> [4] Learning to Contrast the Counterfactual Samples for Robust Visual Question Answering
>
> [5] Representation Learning with Contrastive Predictive Coding
>
> Q2: Do you know what percentage of the training data is not well captured by an (event, relation, ?) query? That is, how often does your decomposition fail?
>
> Response: Firstly, all warm-up questions, such as “What has already happened”, cannot be well captured by a query. Hence we add a special event “current time” as e_q as shown in section 3.2.1. Secondly, for the rest of the questions, only less than 3% questions cannot be separated. Here the same process used in the warm-up questions is taken to deal with these unseparated questions.
>
> Q3: In section 3.2.1, it's not clear what exactly is being averaged to compute v_q - all of the words in the question?
>
> Response: Each question event e_q usually contains multiple words. For example, the question event includes three words {'Kilpatrick', 'Robyn', 'said'} for the question “what happened after Robyn Kilpatrick said?” Hence we average the representations of these words as the v_q.
>
> Q4: Section 4.4 - t-SNE is generally not very informative, and I disagree with the text in this section that says that the OTR model has better clustering than the baseline. It does not seem nearly so clear cut to me as is stated in the text. I think this section would be better to just be removed from the paper. It doesn't really add anything that's scientifically useful.
>
> Response: I couldn't agree with you more. The t-SNE figures are usually used for the visualization without providing much information. Our purpose here is to show the embeddings of open temporal relations could be well separated. We will modify this section based on your suggestion.

---

### Official Review · Reviewer_udhY · 2021-07-21
**Solid study on open temporal relations for QA, but the empirical result seems marginal**

**Rating:** 6
**Confidence:** 3

**Review:**

**Summary**
Authors introduce a temporal question answering (TQA) model called OTR-QA, which decomposes a given question into a question event and a temporal relation. Their main contribution is to learn open temporal relations that do not require a predefined set of relations. The architecture is simple and trained with multiple hyperplanes. Answer event prediction loss is further assisted by two auxiliary losses (event detection loss and contrastive loss). Problem definition is straightforward (i.e., extract all the answer entities given the question entity and the relation) and the model is evaluated on TORQUE (Ning et al., 2020). Compared to the RoBERTa-large baseline by Ning et al., 2020, OTR-QA achieves +1.1% F1, -0.2% EM, and +3% C score (consistency). Authors also show that their model is more data-efficient and modeling the open temporal relations is indeed the key to achieve strong performance.

**Strengths**
* The paper is well-structured and easy to follow.
* The ablation study in Table 3 is helpful to understand why their model works well.
* Their model shows significant improvement over RoBERTA-base in the low-resource setting.

**Weaknesses**
* The performance of the RoBERTa-large baseline in the TORQUE leaderboard is (F1=0.7562, EM=0.5076, C=0.3691), which largely closes the gap with OTR-QA, especially in C score. I understand that authors reported the RoBERT-large performance from Ning et al., 2020, but it would be nice if authors can explain why this happens and report the performance of OTR-QA evaluated on the official leaderboard.
* Their main result (Table 3) might have omitted comparison with previous works on TORQUE (e.g., DEER: A Data Efficient Language Model for Event Temporal Reasoning. Han et al., 2020) but only includes the baselines provided by Ning et al., 2020. It would be better to include them to put their numbers in context.
* In Figure 3, it is not very clear that OTR-QA has better representations of the relations and the color scheme is also confusing.

Overall, I find the paper interesting and clear, but the improvement over previous models seems marginal. It would be nice if authors can provide clarification on these weaknesses.

---

> ### Author Response · Authors · 2021-07-28
> **Thanks for your review and thoughtful suggestions!**
>
> Thank you for the thoughtful suggestions, which will make the paper better. All comments have been carefully taken into consideration. We give the following responses to clarify the proposed method and answer your concerns.
>
> Q1: The performance of the RoBERTa-large baseline in the TORQUE leaderboard is (F1=0.7562, EM=0.5076, C=0.3691), which largely closes the gap with OTR-QA, especially in C score. I understand that authors reported the RoBERT-large performance from Ning et al., 2020, but it would be nice if authors can explain why this happens and report the performance of OTR-QA evaluated on the official leaderboard.
>
> Response: Thanks for your question. Firstly, we reported the numbers of RoBERTa-large from paper since other numbers are also coming from this paper, which keeps the consistency. Secondly, we only achieved the similar performance presented in the paper instead of the performance of the leaderboard by our experiments. Thirdly, another paper DEER [Han et al., 2020] from one of the authors who proposed the TORQUE challenge also reported the numbers used in the paper instead of leaderboard.
>
> Q2: Their main result (Table 3) might have omitted comparison with previous works on TORQUE (e.g., DEER: A Data Efficient Language Model for Event Temporal Reasoning. Han et al., 2020) but only includes the baselines provided by Ning et al., 2020. It would be better to include them to put their numbers in context.
>
> Response: Thanks for your suggestion. Our paper contains an explanation about this concern in the footnote of page 8:
> “We do not include the model and results from [Han et al., 2020] because it used different training data in addition to TORQUE.”
> The DEER [Han et al., 2020] created large-scale masked samples as distant supervision signals. It takes advantage of the 20-year’s New York Times news articles with 10 million sentence passages for training. So it is hard to directly compare our OTR-QA to the DEER method.
> The performance of the DEER [Han et al., 2020] with much more training data is F1 76.1, EM 51.6 and C 36.8, which is similar to us. Thus these results suggest that our approach is data efficient since we haven’t used additional data but achieved the SOTA performance. Some comparisons will be added in the final version.
>
> Q3: In Figure 3, it is not very clear that OTR-QA has better representations of the relations and the color scheme is also confusing.
>
> Response: Our purpose using the Figure 3 is to show OTR-QA could learn better embeddings of open temporal relations. Here a better representation means that the similar relations are grouped together tightly and different relations are well separated. The cluster w.r.t. OTR-QA (right sub-figure) is much tighter and well separated. We will update these figures with a new color scheme with high separability. More explanations will be added in the final version.

---

### Official Review · Reviewer_UY5T · 2021-07-21
**Intuitive method with positive results.**

**Rating:** 7
**Confidence:** 4

**Review:**

Strengths:
- Clear, well explained motivations
- Positive results compared to baseline RoBERTa methods
- Ablation study to show the benefits of the proposed model

Weaknesses
- Gains are not huge for EM and F1, especially for the test set

This paper presents an intuitive technique with positive results, I am a bit concerned that the observed gains might be due to noise/randomness, but at least the consistency score and low-data results seems to have been improved by a substantial margin.

Question:
- Were the results averaged over multiple trials? It would help substantially to show averaged results for at least the validation set, especially since some of the margins are small
- Are these results statistically significant given the size of the test/validation sets?
-  Is the MLP from equation 7 used at test time? If not, it seems like that would limit the benefit of the contrastive loss. Is there a way to share parameters between the MLP in the contrastive loss and the logistic regression model in the event detection loss so the contrastive loss more directly effects the model's output?

Presentation:
- Equation 2: Should there be commas between the h and i in the W subscript?
- Table 2: 50.9 is bolded even though it is not the highest score in that column, is that intentional?
- Equation 8: “is a binary event indicator (1 or 0) if” -> “is a binary event indicator (1 or 0) and”

---

> ### Author Response · Authors · 2021-07-28
> **Thanks for your review and thoughtful suggestions!**
>
> We thank the reviewer for the constructive suggestions regarding the evaluation of our proposed method. We answer these questions in the following.
>
> Q1:  I am a bit concerned that the observed gains might be due to noise/randomness, but at least the consistency score and low-data results seems to have been improved by a substantial margin. Were the results averaged over multiple trials? Are these results statistically significant given the size of the test/validation sets?
>
> Response: First we want to point out that our OTR-QA has two advantages: 1) our model learns context-agnostic, free-text-based temporal relation representations, which leads to higher data efficiency. As you mentioned, our model has a significant improvement under the low-source setting, such as 5.3% and 20.4% gains on F1 using 50% and 10% training data in Table 2. 2) OTR-QA models the differences in temporal relations with a contrastive loss function so that the consistency score has been improved by a substantial margin, such as a gain of 9.4% on the test set under the RoBERTa-base setting. We appreciate you noticed these two main improvements in the experiments. All improvements are due to reformulating the temporal QA task to an open temporal relation extraction task rather than noise.
>
> Secondly, the performance gain of our model is comparable to the DEER method, which is proposed in the paper of “DEER: A Data Efficient Language Model for Event Temporal Reasoning. Han et al., 2020” in the footnote of page 8. This approach not only uses TORQUE data but also takes extra 10 million sentence passages for training the model.  Even though DEER uses more data, the improvement is still similar to us (our F1 and C scores are slightly better). So the improvement under this challenge task is enough to show the benefit of our model. What’s more, this comparison further proves our OTR-QA is more data-efficient since we haven’t used additional data but achieved the SOTA performance.
>
> Thirdly, our validation metrics are averaged over multiple trials. All standard deviations of our OTR-QA are less than 0.01. In addition, we have added a t-test on the validation set by comparing our method to baseline with RoBERTa-base in terms of F1 on multiple trials. The p-value is 0.016, which proves statistically significant.
>
> Q2:  Is the MLP from equation 7 used at test time? If not, it seems like that would limit the benefit of the contrastive loss. Is there a way to share parameters between the MLP in the contrastive loss and the logistic regression model in the event detection loss so the contrastive loss more directly affects the model's output?
>
> Response: This is a great suggestion. It is true that we use the MLP during training instead of testing. We appreciate your suggestion using shared parameters to take the benefit of contrastive loss. One possible way is to share the parameters for scoring answer events in Eq. 4  to do the same computation as MLP in Eq. 7.
>
> Text Revisions:
> Thanks for checking our paper carefully, we have fixed these in the revised version.

---

### Decision · Program_Chairs · 2021-08-18

**Decision:**

Accept

**Comment:**

The paper proposes to reformulate temporal question answering as open temporal relation extraction task. All reviewers agree that this is a fine paper which executes the idea well and evaluates it thoroughly, including ablation experiments. A good and timely piece of work. The approach works well for temporal QA without the need of explicit timestamps: the experiments show that the decomposition generalize better and models better different temporal relations with the contrastive loss.